# An Overview of the Photocatalytic Water Splitting over Suspended Particles

Muhammad Amtiaz Nadeem [1], Mohd Adnan Khan [2], Ahmed Abdeslam Ziani [1] and Hicham Idriss [1,*]

1    SABIC Technology Center, King Abdullah University of Science and Technology, Thuwal 23955, Saudi Arabia; nadeemmi@sabic.com (M.A.N.); ziania@sabic.com (A.A.Z.)
2    Department of Chemical and Petroleum Engineering, University of Calgary, NW Calgary, AB 2500, Canada; mohd.khan1@ucalgary.ca
*    Correspondence: idrissh@sabic.com or h.idriss@ucl.ac.uk; Tel.: +966-12-28524444

**Abstract:** The conversion of solar to chemical energy is one of the central processes considered in the emerging renewable energy economy. Hydrogen production from water splitting over particulate semiconductor catalysts has often been proposed as a simple and a cost-effective method for large-scale production. In this review, we summarize the basic concepts of the overall water splitting (in the absence of sacrificial agents) using particulate photocatalysts, with a focus on their synthetic methods and the role of the so-called "co-catalysts". Then, a focus is then given on improving light absorption in which the Z-scheme concept and the overall system efficiency are discussed. A section on reactor design and cost of the overall technology is given, where the possibility of the different technologies to be deployed at a commercial scale and the considerable challenges ahead are discussed. To date, the highest reported efficiency of any of these systems is at least one order of magnitude lower than that deserving consideration for practical applications.

**Keywords:** overall water splitting; hydrogen production; UV and Vis light responsive catalysts; Z-scheme; mass transfer; future R&D perspective





## 1. Introduction

The increasing use of fossil fuels is the cause of considerable concern due to the generation of greenhouse gases, and their link to global warming. The last few years have been recorded as the hottest ever on the planet [1]. There is a need to find alternative sustainable sources and vectors of energy, ideally with zero carbon emissions. The sun is an inexhaustible and clean source of energy. It provides energy in one hour ($432 \times 10^{18}$ J) close to our annual demand ($575 \times 10^{18}$ J) [2–4]. However, its low power flux-density ($1 \ \text{kW m}^{-2}$ at maximum) coupled with its intermittent nature dictates the need to develop technologies for solar energy storage [3–5].

Hydrogen is one of the best options to store solar energy since it has the highest gravimetric energy density ($142 \ \text{MJ kg}^{-1}$) of all known compounds and produces zero carbon upon combustion. It is also a reactant in industrially important chemical processes such as CO hydrogenation to methanol and ammonia production (Haber-Bosch reaction) [6,7]. It is largely produced via steam methane reforming (SMR) and coal gasification, both result in significant amounts of $CO_2$ emission [8]. Although in small quantities (2–3% of total hydrogen production), it is also produced by electrolysis of water. The process can be made renewable if the electricity used is obtained from renewable resources such as solar or wind [9,10]. The economy of the process is driven by electricity costs and despite decreasing renewable electricity costs, hydrogen made via electrolysis is still expensive when compared to that produced by SMR [11,12].

Photocatalytic systems that drive the water splitting reaction on the surface of semiconductor nanoparticles using solar energy could be a promising alternative. This method has the advantage of using low cost photocatalysts when compared to photovoltaic cells, along

with simplicity of reactor design [13–15]. Various reports on the techno-economic analysis (TEA) of these systems propose a target of ~10% solar-to-hydrogen energy conversion efficiency (STH) for the process to be economically viable [13,14]. However, these reports are based on many questionable assumptions in addition to several challenges that need to be addressed before further considerations for their commercial/economic viability. The first major hurdle is to improve the efficiency of photocatalysts itself for overall water splitting (OWS). After 50 years of research or so, the most efficient photocatalyst demonstrated a STH of ~1% (a performance that needs to be reproduced by other groups), which is far from the minimum proposed target of 10% [16]. The second subsequent hurdle is the demonstration of long-term stability for these photocatalysts which otherwise would have a significant effect on the economics of the process. In addition, the development of a technology for the separation and compression of molecular hydrogen and oxygen from a photocatalytic reactor is a daunting challenge. Without addressing these challenges, studies on the commercial and economic viability of photocatalytic hydrogen production would be irrelevant.

To facilitate scientific progress in this field, it is important to take a critical look and understand the underlying principles governing fundamental processes such as light absorption, charge carrier generation and their recombination, charge transfer, redox reactions and mass transfer limitations that in combination dictate the overall efficiency of a photocatalytic system. In this regard we present a review of such systems that attempted to improve the overall process efficiency by addressing the fundamental steps involved. The review starts by presenting the basic concepts involved in the overall water splitting reaction using single component photocatalysts through one-step photoexcitation. This is followed by summarizing the most used strategies for the design and synthesis of these materials. Subsequently, approaches taken to improve light absorption, treating UV and visible light absorbing materials separately are presented. In the later sections of the review, approaches to reduce charge carrier recombination and few reports addressing mass transfer limitation and the prevention of $H_2/O_2$ back-reaction are discussed [16–19]. The review concludes with a brief techno-economic analysis taking into consideration possible reactor configurations for photocatalytic hydrogen production and future perspectives.

## 2. Fundamental Processes in Photocatalytic Overall Water Splitting

The first report on water splitting using light radiations considered a photoelectrochemical cell with a $TiO_2$ photoanode [20]. This demonstration led to numerous efforts for photocatalytic water splitting on powder semiconductors in the late 20th century [20–27]. Metal-loaded semiconductor (such as $Pt/TiO_2$) can be considered as a short-circuit photo-electro-chemical (PEC) cell that offers both oxidation and reduction centers on the same catalyst (see Figure 1) [28]. A photocatalytic reaction is initiated by the absorption of photons with energy higher than the bandgap ($E_g$) of a semiconductor material (step 1). Thermodynamically, the bandgap energy should be larger than 1.23 V and band positions below 1.23V vs Normal Hydrogen Electrode (NHE) to abstract electrons from $O^{2-}$ anions of water and above 0.0V vs NHE to reduce $H^+$ ions of water. Following light absorption, electrons are transferred to the conduction band (CB) with simultaneous generation of holes at the valence band (VB) (step 2). The recombination of photogenerated charge carriers in the bulk or on the surface (step 3) competes with their diffusion to the surface to carry the redox reactions (steps 4 and 4'). The presence of metal particles with higher work function than the oxide semiconductor may pump excited electrons to increase the rate of charge separation and to reduce $H^+$ ions to atomic hydrogen that then recombine to make $H_2$. At the energy level of the valence band, electrons are abstracted from $O^{2-}$ anions of water to close the cycle. Ultimately, the reaction products desorb from the catalyst surface and are transferred to the medium to complete the overall process.

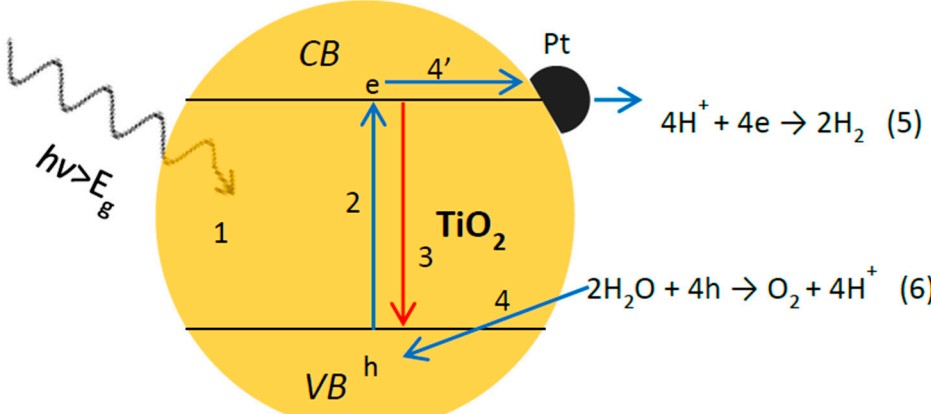

**Figure 1.** Schematic representation of photocatalytic overall water splitting on a metal-loaded semiconductor (such as Pt/TiO$_2$) particle system: (1) light absorption, (2) electron excitation from the VB to the CB, (3) e-h recombination, (4) electron transfer from O$^{2-}$ to the VB, (4') electron transfer from the metal to H$^+$ ions reduction and (5 and 6) H$_2$ and O$_2$ production.

## 3. Design and Synthesis of Particulate Photocatalytic Systems

Currently, photocatalytic water splitting research is focused on two approaches; one-step excitation using a single particulate photocatalyst [29–39] and a two-step excitation using coupled two individual particulate photocatalysts with an electron transfer mediator, known as a Z-scheme process [17,40–48]. The first approach requires the development of a particulate photocatalyst that is capable of being excited by light and efficiently performs all the steps as described in Figure 1 on the same surface (since it is composed of multiple phases). This implies many daunting requirements including a suitable band gap energy and bands positions with sufficient driving potential, efficient separation and transport of electron–hole pairs and photo-corrosion resistance for a given semiconductor particle [29–39]. The second approach requires two (ideally) visible-light-driven photocatalysts for each of the half reactions (either H$^+$ reduction or OH$^-$ oxidation). The semiconductor band positions, redox potentials of water and that of the mediator used must be aligned in such a way to allow electron transfer from the oxygen to that of the hydrogen evolution centers [17,40–48]. It also requires that the transport and the catalyst to be chemically inert (so irreversible adsorption (poisoning by the charge transfer mediators) does not occur for example). Further details about both approaches are presented later in the review. Consequently, the electronic properties and materials design are the main challenges.

Particulate photocatalytic materials that are able to perform a water splitting reaction can be divided into two categories; the metal oxide and (oxy-) nitrides with metal particles on top, and metal-free catalysts. For the first category, which is the most widely studied, many methods of preparation have been used. The most common one is the solid-state reaction synthesis used for di- or multi-metallic oxides. For example, Ba doped La$_2$Ti$_2$O$_7$, NaTaO$_3$, SrTiO$_3$:(Rh, Sb), BiVO$_4$:(In, Mo), Bi$_{1-x}$In$_x$V$_{1-x}$Mo$_x$O$_4$ and BiYWO$_6$ photocatalysts have been prepared [49–53]. To prepare them, commercial metal oxides and alkali carbonates are mixed in a stoichiometric ratio followed by mechanical grinding in a solvent such as ethanol and dried in an oven. In general, the dried powder, in general, is pressed into the form of a pellet followed by calcination in the 800–1000 °C temperature range for many hours to obtain the appropriate composition and phase. The flux or molten salt methods have also been used for catalyst preparation. For example, SrTiO$_3$:Al catalyst is prepared by mixing SrTiO$_3$ and Al$_2$O$_3$ in the presence of SrCl$_2$ flux in an appropriate molar ratio followed by heating at 1100 °C in air [15,54]. The flux method is particularly important to reduce thermal strains during preparation and may reduce the defect densities in the final product.

In the case of monometallic oxides, they can be prepared under controlled atmosphere using a variety of methods. For example, $Co_3O_4$ powder (size <10 μm,) decomposes into CoO powder in a quartz tube furnace at 1000 °C in an Ar environment [55]. $Ga_2O_3$ is prepared from $Ga(NO_3)_3.xH_2O$ using the ammonia precipitation method followed by calcination under oxygen atmosphere [56,57]. Most of the above methods produce metal oxide catalysts active under UV irradiation. These as prepared oxides can be heated under ammonia flux at high temperature to convert them into (oxy-) nitrides photocatalysts, for example $LaMg_{1/3}Ta_{2/3}O_2N$, $TaON:ZrO_2$, $CaTaO_2N$, $LaSc_{0.5}Ta_{0.5}O_2N$, $Ta_3N_5$ and $(Zn_{0.18}Ga_{0.82})(N_{0.82}O_{0.18})$ which are active under visible irradiation [58–64]. One common issue with all the above conventional preparation methods is that it is hard to obtain pure and defect free materials. Other methods include plasma-assisted molecular-beam epitaxy, for example, to make GaN:Mg/InGaN:Mg [62]. For metal free catalysts, they are mainly based on graphitic carbon nitride (g-$C_3N_4$). Urea is used as a source of carbon and nitrogen whose appropriate calcination for many hours resulted in g-$C_3N_4$ [65]. To further improve the performance, g-$C_3N_4$ can be mixed with carbon dots (C-dots) [66]. The latter can be synthesized by a typical electrochemical exfoliation followed by hydrothermal treatment with ammonia.

In most cases, a photocatalyst requires a metal in addition to a semiconductor. This is commonly called a "co-catalyst" to help improve the kinetics of the reaction, by facilitating charge transfer between semiconductor and the reactants. The word co-catalyst is misleading, most heterogeneous catalysts are composed of multiple elements and phases and therefore there is no need to label one part as a catalyst and another as a co-catalyst. Wet impregnation and photo-deposition are the two most commonly used methods to load these metals. Among these, the former method is the most widely used, simply because the latter does not provide a better dispersion nor a more homogenous particle size distribution. For example, in the case of $Rh_{2-y}Cr_yO_3$ supported on $SrTiO_3:Al$, an aqueous solution containing the needed concentrations of $Na_3RhCl_6$ and $Cr(NO_3)_3$, in addition to the semiconductor, is followed by evaporation [15,54] and then calcination at 300 °C. A source of light irradiation is used for the photodeposition of metals. For example, consider $RuO_2$ photo-deposition on $Bi_{1-x}In_xV_{1-x}Mo_xO_4$ [52,53], where the powder (semiconductor) and $RuCl_3.6H_2O$ precursor are dispersed in methanol (hole scavenger)-water mixture under light irradiation to obtain the final metal loaded photocatalyst. Table 1 presents a summary of the main methods for photocatalyst preparation used for the water splitting reaction.

**Table 1.** Major photocatalysts synthesis methods for overall water splitting applications.

| Photocatalysts | | Preparation Method | | Examples | | Cocatalyst Loading | | Examples |
|---|---|---|---|---|---|---|---|---|
| Metal oxides | → | Molten salt (Flux) Solid state reactions Ammonia precipitation | → | $SrTiO_3:Al$, $SrTiO_3:Rh,Sb$, $La_2Ti_2O_7:Ba$, $NaTaO_3$, $Ga_2O_3:Zn$, $BiYWO_6$, $Bi_{1-x}In_xV_{1-x}Mo_xO_4$ | → | Impregnation Photodeposition | → | $NiO_x$, $CoO_x$, $IrO_2$, $RuO_2$, $Rh_{2-y}Cr_yO_3$ |
| | | Calcination under controlled atmosphere | → | $M_xO_y$ | | | | |
| Metal (oxy) nitrides | → | Thermal nitridation of metal oxides using $NH_3$ | → | $Ge_3N_4$, $TaON:ZrO_2$, $(Zn_{0.18}Ga_{0.82})(N_{0.82}O_{0.18})$, $LaMg_{1/3}Ta_{2/3}O_2N$, $CaTaO_2N$, $Ta_3N_5$, $LaSc_xTa_{1-x}O_{1-2x}N_{2-2x}$, GaN:Mg/InGaN:Mg | → | Impregnation Photodeposition | → | $RuO_2$, $Rh_{2-y}Cr_yO_3$ |
| Metal-free photocatalysts | → | Thermal polymerization Electrochemical | → | g-$C_3N_4$, C-dot/g-$C_3N_4$ | | | | |

## 4. Improving Light Absorption

### 4.1. UV Light Photocatalysts

The UV-light excited photocatalysts are among the earliest ones studied since the first report on the water splitting reaction using a $TiO_2$ photoelectrode in 1972 [67]. These are mainly oxides of group IV-VI transition metals with $d^0$ ($Ti^{4+}$, $Zr^{4+}$, $Nb^{5+}$, $Ta^{5+}$ and $W^{6+}$) [68,69] and group III-V p-block elements with $d^{10}$ ($Ga^{3+}$, $In^{3+}$, $Ge^{4+}$, $Sn^{4+}$ and

$Sb^{5+}$) [70–72] electronic configurations of their cations. While O2p orbitals make valence band for both types, the empty d-orbitals of $d^0$ and hybridized sp-orbitals of $d^{10}$ oxide materials form conduction bands. They possess a large bandgap (>3 eV) mainly due to the electrochemical potential of O2p orbitals located near 3.0 V (vs. NHE at pH = 0) [73]. There are very few reports on UV-light responsive non-oxide catalysts [58]. Metal oxide catalysts such as Al-doped $SrTiO_3$ [54] or Zn-doped $Ga_2O_3$ show noticeable one-step photocatalytic water splitting activity reaching 71% apparent quantum yield (AQY) at 254 nm of light irradiation [57]. Apparent quantum yield is defined as the ratio of the number of reacted electrons or holes with the reactants to the number of incident photons on the photocatalysts. Among metal oxide catalysts, $TiO_2$ is the most commonly studied due to its superior photo-stability, nontoxicity, high activity for dye degradation [74] and hydrogen production using sacrificial agents (upon UV radiation λ <387 nm) [29–34]. There are few reports about overall water splitting using $TiO_2$ as well; however, its activity is below par as compared to other oxide catalysts [22,23,25,75,76]. Detailed reviews on $TiO_2$-based materials and their photocatalytic performances can be found elsewhere [75,77,78]. The hydrogen and oxygen evolution activity under UV radiation and the apparent quantum yield of some of these selected materials are given in Table 2.

**Table 2.** Hydrogen and oxygen evolution activity of $d^0$ and $d^{10}$ metal oxide particulate catalysts under UV light irradiation.

| Semiconductor | Metal/Metal Oxide (wt.%) | $E_g$ (eV) | $H_2$ Rate (mmolh$^{-1}$) | $O_2$ Rate (mmolh$^{-1}$) | AQY (%) | Ref. |
|---|---|---|---|---|---|---|
| $La_2Ti_2O_7$:Ba(8.0 mol %) | Ni (2.0) | 3.2 | 5 | 2.5 | 50 (*not given*) | [49] |
| $SrTiO_3$:Al(0.1 mol %) | $Rh_xCr_yO_3$ (Rh = 0.1, Cr = 0.1) | 3.2 | 1.4 | 0.7 | 56 at 365 nm | [15,54] |
| $SrTiO_3$:Al(0.1 mol %) | $MoO_y/RhCrO_x$ (Mo = 0.03, Rh = 0.1, Cr = 0.1) | 3.2 | 1.8 | 0.9 | 69 at 365 nm | [54] |
| $SrTiO_3$:Rh,Sb(0.5 & 2.0 wt.%) | $IrO_2$ (3.0) | 3.2 | 4.4 | 1.9 | 0.1 at 420 nm | [50] |
| $NaTaO_3$ | NiO (0.05) | 4.0 | 3.4 | 1.6 | 20 at 270 nm | [51] |
| $Ga_2O_3$:Zn(1.0 mol %) | Ni (1.0) | 4.4 | 4.1 | 2.2 | 20 at 270 nm | [56] |
| $Ga_2O_3$:Zn(3.0 mol %) | $Rh_xCr_yO_3$ (Rh = 0.5, Cr = 1.5) | 4.4 | 3.2 | 1.6 | 71 at 254 nm | [57] |
| $Ge_3N_4$ | $RuO_2$ (1.0) | 3.8 | 0.2 | 0.1 | 9 at 300 nm | [58] |

An enhancement in the apparent quantum yield in a semiconductor photocatalyst is generally achieved after loading metal (metal oxide) for $H_2$ ($O_2$) production. They improve photocatalytic activity by promoting charge separation by extracting photo-generated electrons and holes. They may also inhibit the photodegradation of a photocatalyst due to the kinetics promotion of catalytic reactions at the surface, thus, decreasing the probability of charge carriers' reaction within the semiconductor itself [79,80]. As for most multicomponent heterogeneous catalytic materials, attention needs to be given to the energy levels of each component with respect to the Fermi level, which is also a function of their surface and bulk atomic structure. This is to permit the formation of the desirable junction (an Ohmic or a Schottky-type contact) allowing charges to flow in the needed direction at the interface. Transition metals, in particular precious metals, such as Pt [24,25], Rh [22], Ru [81], and Au [82,83] and also non-precious metals such as Cu [84] and Ni [23] have been studied to promote the hydrogen evolution reaction. On the other hand, $CoO_x$ [85,86], $FeO_x$ [87], $MnO_x$ [87], $RuO_2$ [88] and $IrO_2$ [89] promote the oxygen evolution reaction. For the majority of water splitting photocatalysts, the loading of a reduction center is obligatory due to an insufficient over-potential for $H^+$ reduction as compared to $H_2O$ oxidation. In particular, the role of Ni/NiO is in particular noteworthy since Ni captures electrons from the CB of the semiconductor (to reduce hydrogen ions) and NiO captures electrons from $O^{2-}$ anions of water [49,51,56,90,91].

### 4.2. Visible Light Photocatalysts

As noted above, one of the main issues with oxide catalysts is their high band gap energy and the UV portion of the solar spectrum is up to 4% or so only [92]. To enable visible light utilization, the valence band level of these semiconductors should be shifted upward without changing the conduction band position. One approach to achieve this is to use oxynitrides to take advantage from N2p states that lie at a more negative potential than O2p states (Table 3) [37–39]. In general, Ta- and Ga-based oxynitrides with partial La or Ga substitution with other metals (Zn, La, Nb, Ca, Ba, Mg, Sr, and Sc), show activities under visible light irradiation. For example, $LaMg_xTa_{1-x}O_{1+3x}N_{2-3x}$ and $(Ga_{1-x}Zn_x)(N_{1-x}O_x)$ are representative oxynitride photocatalysts capable of water splitting under visible-light irradiation [37,93]. While the formula $LaMg_xTa_{1-x}O_{1+3x}N_{2-3x}$ represents a solid solutions of $LaTaON_2$ and $LaMg_{2/3}Ta_{1/3}O_3$, $(Ga_{1-x}Zn_x)(N_{1-x}O_x)$ is a solid solutions of GaN and ZnO [39,64]. The compounds $LaSc_xTa_{1-x}O_{1+2x}N_{2-2x}$ ($x \geq 0.5$) and $CaTaO_2N$ are also capable of driving the water splitting reaction and both are related to $LaMg_xTa_{1-x}O_{1+3x}N_{2-3x}$. In these catalysts La or Ta sites in the perovskite structure are replaced by Ca or Sc that alter O/N ratios due to charge compensation, which in turn results in a valence band energy-level shift [63,64]. The conduction band minimum and valence band maximum for these materials are invariably composed of Ta5d and N2p orbitals, respectively. Recently, overall water splitting was made possible on epitaxially grown $Ta_3N_5$ nanorods. The activity was attributed mainly to the elimination of crystalline defects in the photocatalyst [94,95].

**Table 3.** Hydrogen and oxygen evolution activity of photocatalysts under visible light irradiation.

| Semiconductor | Metal Oxide (wt.% Unless Indicated) | Eg (eV) | H$_2$ Rate ($\mu$molh$^{-1}$) | O$_2$ Rate ($\mu$molh$^{-1}$) | AQY (%) | Ref. |
|---|---|---|---|---|---|---|
| $Bi_{1-x}In_xV_{1-x}Mo_xO_4$ | $RuO_2(3.0)$ | 2.5 | 17 | 7.8 | 3.2 at 420 nm | [52] |
| $BiYWO_6$ | $RuO_2(1.0)$ | 2.7 | 4.1 | 1.8 | 0.17 at 420 nm | [53] |
| $LaMg_{1/3}Ta_{2/3}O_2N$ | $RhCrO_x$ (Rh = 0.5 Cr = 0.5) | - | 22 | 11 | 0.18 at 440 nm | [59] |
| $TaON{:}ZrO_2$ (Zr/Ta = 0.1) | $RuO_x/Cr_2O_3/IrO_2$ (Ru = 3.0, Cr = 2.5, Ir/Ta = 0.04) | 2.5 | $6.7 \times 10^{-3}$ | $2.3 \times 10^{-3}$ | <0.1 at 420 nm | [60] |
| CoO | - | 2.6 | 1785 | 848 | 5% (STH) | [55] |
| $(Zn_{0\cdot18}Ga_{0\cdot82})(N_{0\cdot82}O_{0\cdot18})$ | $Rh_{2-y}Cr_yO_3$ (Rh = 2.5, Cr = 2.0) | 2.64 | 927 | 460 | 5.9 at 420 nm | [61] |
| GaN:Mg/InGaN (grown using MBE) | $Rh/Cr_2O_3$ (Not applicable) | 2.22 | 38 | 21 | 12.3 at 400 nm | [62] |
| $CaTaO_2N$ | $RhCrO_y$ (Rh = 0.5, Cr = 0.5) | 2.43 | $14 \times 10^{-2}$ | $7 \times 10^{-2}$ | 0.003 at 440 nm | [63] |
| $LaSc_{0.5}Ta_{0.5}O_2N$ | $RhCrO_y$ (Rh = 0.5, Cr = 0.5) | 2.1 | $2.4 \times 10^{-3}$ | $1.2 \times 10^{-3}$ | - | [64] |
| $Ta_3N_5/KTaO_3$ ($Ta_3N_5$ = 1.4 wt.%) | $Rh/Cr_2O_3$ (Rh = 0.002, Cr = 0.004) | 2.1 | $6 \times 10^{-3}$ | $3 \times 10^{-3}$ | 0.25 at 400 nm | [94] |
| $g-C_3N_4$ | $CoO_x$ (1.0) | 2.8 | 8.5 | 3.5 | 0.3 at 405 nm | [65] |
| $C_3N_4/C$-dots | - | 2.74 | 46 | - | 16 at 420 nm | [66] |

One of the main issues often debated for nitrogen-doped photocatalysts is their stability. In most of the cases, their photocatalytic activity is reported for a few hours (often less than 20 h). While some of these catalysts are reported to be stable on this time scale, most show clear decrease in their activities implying that any of them may not be stable over extended periods given their similar chemistry [37,60,79,93,94]. However, it is stunning that so far after decades of work no standard has been put in place in order to ensure that at least each catalytic site is used more than one. Turn over Numbers (TON) like terms can be introduced. For example for a reaction requiring GaN as a catalyst the number of hydrogen molecules should be compared to the number of Ga atoms in the catalyst so as to ensure that the reaction is catalytic for at least two cycles. There is no strict heterogeneous catalytic definition of TON. In this case, one can define TON as the total number of reaction product divided by the total number of metal cations of the catalyst.

Apart from nitrogen doping, the incorporation of $d^n$-transition (n= 1–10) metal cations also results in visible-light absorption by forming energy states above O2p states, but these cations also tend to function as recombination centers thus showing overall little success [73,96]. A relatively successful example of this strategy is Rh, Sb-doped $SrTiO_3$ [97]. The visible light absorption results due to electron transition from the Rh4d state to the conduction band that is primarily composed of Ti3d orbitals. While Rh doping induces hydrogen evolution ability, Sb co-doping can induce oxygen evolution activity, albeit with reduced capacity for hydrogen evolution. Therefore, an adjustment in the Rh/Sb ratio to achieve an optimal balance between hydrogen and oxygen evolution was found to be important [50,60,63]. Yet, it is unclear if the reaction is indeed catalytic for these type of catalysts because turnover numbers were not reported in order to ensure that each site is used more than once.

Metal-doped $BiVO_4$ is another example which in its pristine monoclinic phase is unable to drive the water splitting reaction because of its thermodynamically insufficient conduction band minimum potential. However, the co-doping of $BiVO_4$ with In and Mo results in a partial transformation of the pure monoclinic crystal structure to a mixture of monoclinic and tetragonal structures. This leads to an increase in its unit cell volume that results in a negative shift in the conduction band minimum and enables the reaction [52]. Based on this strategy, $BiYWO_4$ was found to function under visible light [53]. Surprisingly, pristine CoO (with a $d^7$ electronic configuration) has been reported to be active even without the presence of a cocatalyst [55]. The highest ever STH for a particulate photocatalyst of 5% was reported for CoO. These results were later supported by others using DFT calculations [98]. However, further work is needed to confirm and clarify this, in particular the stability of CoO nanoparticles under photocatalytic conditions. Graphitic carbon nitride has been reported to function as a photocatalyst in the two-step [99,100] as well as a stand-alone catalyst for a one-step water splitting reaction upon the deposition of a suitable co-catalyst [65]. It was reported that C-nanodots supported $g$-$C_3N_4$ resulted in an active and stable photocatalyst, exhibiting an STH of 2% [66]. However, it is unclear what C-nanodots do in term of catalytic sites or activity however; this result has not been reproduced by others to date.

### 4.3. Z-Scheme: A Two-Step Approach

An alternative way, inspired by natural photosynthesis, to utilize a larger part of the solar spectrum is to use two different photocatalysts with an electron transfer mediator in between (Z-scheme). The Z-scheme needs a large overpotential for the overall reaction due to the accommodation of charge carriers in their respective high-energy states. Another drawback for this system is its requirement of eight photons to complete the wate splitting cycle, whereas a one-step process requires four photons [101]. This difference results in the amounts of hydrogen and oxygen generated equal to half of that produced via one-step photoexcitation system at a given apparent quantum yield and incident light flux [67]. The Z-scheme system only works if the redox potential associated with each photocatalyst component as well as water redox potential are situated with respect to each other as shown in Figures 2 and 3.

The Z-scheme allows flexibility in selecting the semiconductors with small band gaps and therefore opens other routes for visible light utilization. For example, for the $H^+$ ion reduction catalyst, its valence band potential can be located anywhere within the band gap of the oxidation catalysts. This allows the selection of a photocatalyst with conduction band potential much negative that the water reduction potential with much more driving force for the reaction. Similar flexibility is available for the oxidation catalyst. In general, this is not possible in the case of a catalyst used for a one-step reaction. However, care must be taken not to select a catalyst with a band gap wide enough to compromise a visible light absorption range. Based on the type of the redox mediator used, Z-scheme systems can be divided into two types: The Z–scheme with an aqueous redox mediator and with a solid-state mediator. Both types are discussed in the following section.

#### 4.3.1. Z-Scheme with Aqueous Redox Mediator

In such a system, a vectorial electron transfer from the oxidation to reduction catalyst is facilitated by a redox pair without the need of a physical contact between them. (See Figure 2 and Table 4). The reaction may start by the oxidation of water at the valence band of the photocatalyst, while the photo-generated electron is transferred to a mediator (such as $Fe^{3+}$ or $IO_3^-$ ions) to reduce it. Then, the reduced mediator is oxidized back after donating its electron(s) to the conduction band of the hydrogen production catalyst, which subsequently reduces water.

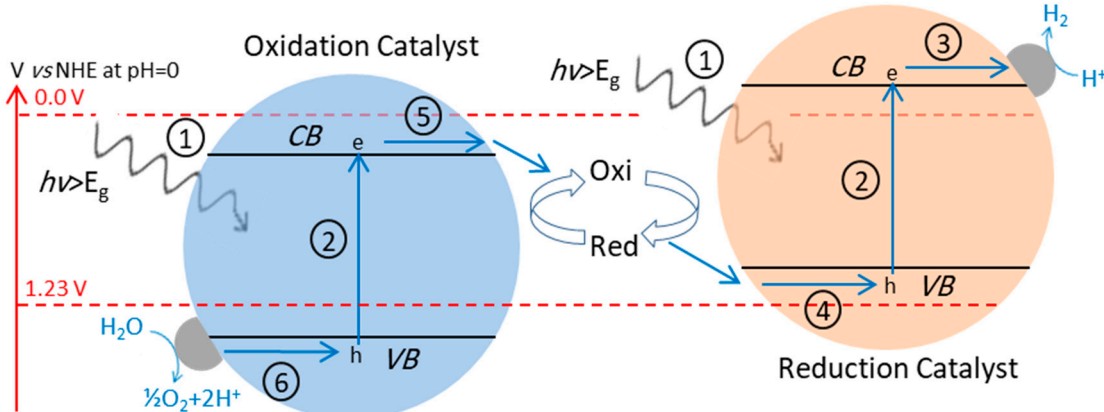

**Figure 2.** Schematic diagram for photocatalytic water splitting using a Z-scheme with an aqueous redox mediator: (1) light absorption, (2) electron excitation from the VB to the CB, (3) electron(s) transfer to the metal cocatalyst for $H^+$ reduction to produce $H_2$, (4) hole(s) transfer to the surface to react with redox mediator, (5) electron(s) transfer to the surface to react with redox mediator and (6) hole transfer to metal oxide cocatalyst to catalyze water oxidation.

The photocatalytic water splitting using $TiO_2$ anatase and rutile as the reduction and oxidation catalysts in 2001 was one of the earliest reports among Z-scheme approaches [102]. The $IO_3^-/I^-$ redox couple, with a redox potential (+0.67 V versus NHE, pH 7), intermediate between those of $H^+/H_2$ and $O_2/H_2O$, was used as an aqueous shuttle. Since then, extensive efforts have been made to use narrow-bandgap semiconductors to achieve overall water splitting driven by visible light. SrTiO3:Rh- based semiconductors and some (oxy)nitrides and (oxy)sulfides, such as $MgTa_2O_{6-x}N_y/TaON$, $C_3N_4$, CdS and $Sm_2Ti_2S_2O_5$, have been the main materials investigated as reduction catalysts [40,100,103–109]. $WO_3$ has been the main focus as an oxidation catalyst due to its activity, stability and ability to absorb visible light (<450 nm) [110–112]. Other materials, include $BiVO_4$, $TiO_2$:Cr/Sb, $Bi_4NbO^8Cl$ and $Ta^3N^5$ [40–43]. Several aqueous redox couples $I^{(5+/-1)}$, $Fe^{(3+/2+)}$, $Co^{(3+/2+)}$ and $Mn^{(3+/2+)}$ have been used as electron mediators [40,104,106,107,113,114].

**Table 4.** Z-scheme-type photocatalysts for water splitting without sacrificial agents.

| H2 Photocatalyst (wt. % Unless Indicated) | O2 Photocatalyst (wt.%) | Mediator | H2 Rate (mmolh$^{-1}$) | O2 Rate (mmolh$^{-1}$) | AQY (%) | Ref. |
|---|---|---|---|---|---|---|
| Pt/SrTiO3:Rh | BiVO4 | $Fe^{3+}/Fe^{2+}$ | 15 | 7.2 | 0.4 at 420 nm | [110] |
| Pt/SrTiO3:Cr/Ta (Pt = 0.3, Cr, Ta = 4.0 mol% each) | PtOx/WO3 (Pt = 0.5) | $IO_3^-/I^-$ | 16 | 8 | 1 at 420 nm | [111] |
| Pt/TaON (Pt = 0.3) | PtOx/WO3 (Pt = 0.5) | $IO_3^-/I^-$_ | 16.5 | 8 | 0.5 at 420 nm | [112] |
| Pt/ZrO2/TaON (Pt = 1.0, Zn/Ta = 0.1) | PtOx/WO3 (Pt = 0.5) | $IO_3^-/I^-$_ | 52 | 27 | 6.3 at 420 nm | [109] |
| Ru/SrTiO3:Rh (Ru = 1.0, Sr:Ti:Rh = 1.1:0.98:0.02) | PtOx/WO3 (Pt = 0.3) | $Fe^{3+}/Fe^{2+}$ | 88 | 44 | 4.2 at 420 nm | [40] |
| Pt/MgTa2O6−xNy/TaON (Pt = 0.4, Mg/Ta = 0.2) | PtOx/WO3 (Pt = 0.45) | $IO_3^-/I^-$ | 108 | 55 | 6.8 at 420 nm | [110] |

In general, a pH adjustment is required to stabilize the redox couple. For example, $I^-/I_3^-$ is stable in acidic where $IO_3^-/I^-$ is stable in basic aqueous media [104,115]. The redox potentials and reactivity of metal cations are tunable through the formation of complexes that also require pH adjustments. For example, in the case of $[Fe(CN)_6]^{3-}/[Fe((CN)_6]^{4-}$, the former requires acidic conditions (with a pH of approximately 2.5) to stabilize the $Fe^{3+}$ ions [40,108]. Various Co-based redox complexes have been reported to function as mediators, although a simple $Co^{3+}/Co^{2+}$ couple is not active [106,108,114].

### 4.3.2. Z-Scheme with Solid-State Electron Mediator

In this approach, charge transfer proceeds either through a solid electron mediator or a physical contact between two different catalysts (see Figure 3).

Among the first reports of a solid-state Z-scheme for an overall water splitting reaction, one was based on a combination of Ru/SrTiO$_3$:Rh and BiVO$_4$ where Rh$^{3+/4+}$ played the role of a mediator [116]. Several other semiconductors such as AgNbO$_3$, TiO$_2$:Rh,Sb and WO$_3$ loaded with suitable oxidation co-catalysts, can also be combined with Ru/SrTiO$_3$:Rh to split water under visible light [41]. A variety of electrical conductors such as Ir, Ag, Au, Rh, Ni, and Pt, reduced graphene oxide and C-dots have been found effective [17,44–48]. The fabrication of a good quality contact of both the catalysts with solid mediator is critical in these systems to allow smooth migration and the recombination of charges at the mediator surface. In these composites, the conduction band of the oxidation catalyst and the valence band of the reduction catalyst should form an Ohmic contact (rather than a Schottky barrier) through the solid mediator to reduce the resistance for the migration of electrons and holes [117,118]. Since noble metals have good inherent photo-corrosion resistance, they have been the preferred choice as electronic mediators [17,44–48].

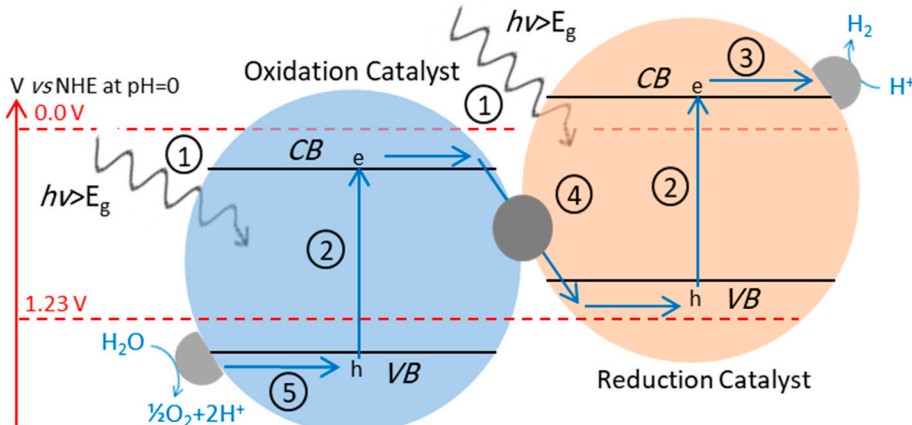

**Figure 3.** Schematic diagram for photocatalytic water splitting using a Z-scheme with a solid-state electron mediator: (1) light absorption, (2) electron excitation from the VB to the CB, (3) electron(s) transfer to the metal co-catalyst for H$^+$ reduction to produce H$_2$, (4) electron(s) transfer from the CB of oxidation catalyst to the VB of reduction catalyst via solid state electron mediator (electron-hole recombination) and (5) hole transfer to a metal oxide cocatalyst to catalyze water oxidation.

In many cases, a direct physical contact is made between the two catalysts to form a simple binary structure that functions as a solid-state electron mediator where charge carriers recombine at the interface of the two semiconductors. This intimate contact can be developed by making use of the electrostatic attractive force obtained by tuning the pH of the reaction medium [47,48,116]. Similar to the Z-scheme with a metal electron mediator, it is highly desirable to have an Ohmic contact with low interfacial resistance between two catalysts to improve the overall charge separation efficiency [117,118]. It has been found that surface defects act as trapping sites for electron–hole recombination by forming a series of intermediate energy levels in the semiconductor bandgaps, which enables the interface between two catalysts to achieve approximately Ohmic contact [119]. This simple structure

has several advantages over a Z-scheme with solid-state mediators. First, the system offers higher long-term stability because of the absence of any photocorrosion associated with a metal mediator. Second, the oxygen and hydrogen recombination reaction is weakened because using the noble metals as an electronic mediator is avoided. Third, a facile and cost-effective fabrication of such systems is possible when compared to those with expensive noble metals. Further details can be found in other reviews dedicated to this topic [101,120].

## 5. Improving Efficiency

### 5.1. Charge Recombination

The charge carrier recombination is largely linked to the material's property where it occurs through various mechanisms within the semiconductor before a chemical reaction at the surface. The charge carrier generation happens at the femto second (fs) time scale [75]. These charge carriers are either transferred to the surface or trapped on a time scale of few tens to few hundreds of fs (50–200 fs) [75,121,122]. In general, the charge carrier recombination in the bulk or on the surface is faster (>20 ns) than the rate of the chemical oxidation/reduction reactions (10–100 $\mu$s) and therefore, it is considered one of the main reasons limiting the process efficiency [75,83,122,123]. According to Leytner and Hupp, 60% of the trapped electron–hole pairs recombine within a timescale of about 25 ns [122,124].

An account of various strategies adopted to reduce charge carrier recombination is given in the following. For example, an increase in crystallinity, to decrease defects such as vacancies and dislocations, most often decreases the charge carrier recombination rate and increases photocatalytic hydrogen production [94]. Pre-calcination of the powders is often an effective mean to increase crystallinity; however, it may also result in lowering the catalyst surface area due to particle sintering. Therefore, it is necessary to balance these two effects to optimize the catalytic activity [125]. In general, for materials with relatively high bulk defect densities, a small particle size minimizes the charge carriers migration distance to the surface reaction centers, thus enhancing their photocatalytic activity [68]. For example, a two-time increase in AQY (from 30% at 260 nm to 56% at 365 nm) on an Al-doped $SrTiO_3$ photocatalyst with a particle size drop from few micrometers to 200 nm has been demonstrated [69,121]. On the other hand, for an undoped semiconductor such as $TiO_2$ the increase in size does not show a similar effect. For example, the rate of photocatalytic hydrogen production on micro and nano size $TiO_2$ was found to be similar when normalized to their surface area [126].

Polymorphic semiconductors are quite common in Nature. An atomically well-matched phase junction can be conveniently fabricated by fine-tuning the phase transformation conditions to enhance charge separation. It has been reported that the water splitting activity of $Ga_2O_3$ is significantly enhanced due to efficient charge separation and transfer across the $\alpha$-$\beta$ phase junction as compared to the individual activities of its constituent (See Figure 4a for details) [127]. The enhanced photocatalytic activity of P25 as compared to individual $TiO_2$ polymorphs is another example [124]. There are some other strategies to decrease charge carrier recombination. For example, Li and co-workers reported the suppression of electron-hole pair recombination on the surface of N-doped $TiO_2$ powders in the presence of a local electric field due to the presence of MgO (111) support [128]. Lately, a few reports have emerged about an increase in photocatalytic activity under an external magnetic field with an objective to decrease charge carrier recombination rates [129–131]. Further work may need to be conducted on this, in particular to see if indeed a magnetic field with the given strength can affect the charge carrier life-time or the charge transfer of a chemical reaction.

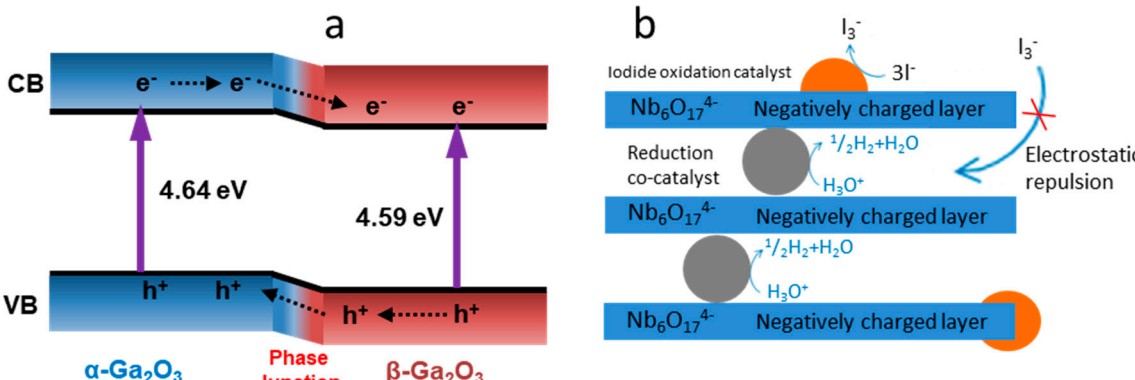

**Figure 4.** (**a**) Band alignment at the α- and β-phase junction of $Ga_2O_3$ for enhanced water splitting activity. Upon irradiation, photogenerated electrons tend to transfer from the α-phase to the β-phase and vice versa for photogenerated holes, which is driven by the potential difference due to their relative band positions. Reproduced with permission from reference [127]. Copyright © (2012), John Wiley and Sons. (**b**) Schematic of reduction catalyst for the Z-scheme with an aqueous redox mediator to avoid backward charge recombination. Gray and orange spheres represent Pt reduction and oxidation cocatalysts, respectively. Reproduced with permission from reference [103]. Copyright © (2011), The Chemical Society of Japan.

In the case of a Z-scheme with an aqueous redox mediator, backward charge transfer to the redox couple is an additional aspect of charge recombination. For example, instead of $IO_3^-/I^-$ reduction and $H_2O$ oxidation, $I^-$ to $IO_3^-$ oxidation and oxygen to $H_2O$ reduction may happen on the oxidation catalyst. One way to circumvent this issue is the selective adsorption of mediator species. It has been found that $IO_3^-$ selectively adsorbs on the surface of $WO_3$, thus preventing the back-reaction. This may also one of the reasons behind $WO_3$ activity to efficiently oxidize water and its preferred choice in Z-scheme system with $IO_3^-/I^-$ mediators [103,109,111–115]. Another strategy to overcome the back-reaction is to spatially separate the oxidation and the reduction sites. For example, in the case of $Pt/K_4Nb_6O_{17}$ reduction photocatalyst, $K_4Nb_6O_{17}$ consists of $K^+$ ion and $Nb_6O_{17}^{4-}$ layers. Pt nanoparticles can be incorporated into the $K^+$ layers through an ion-exchange reaction, whereas oxidation cocatalyst can be loaded on the external surface. This spatial separation of reduction and oxidation sites has been proposed as method to prevent the access of oxidized species ($I_3^-$ or $IO_3^-$) to the Pt particles due to the electrostatic repulsion by the $Nb_6O_{17}^{4-}$ layers while $H_3O^+$ selectively diffuses to reach Pt sites and is reduced (see Figure 4b) [103].

In the case of solid-state Z-scheme systems, the backward flow of electrons and holes to undergo recombination at solid conductors may occur unless a rectification function is induced. Band bending at photocatalyst-solid conductor interfaces may help to rectify the charge transfer direction [16,132]. The backward electron transfer through the solid-state mediator cannot be monitored as easily as in the case of an aqueous electron mediator. This is because the change in aqueous mediator concentration can be measured relatively easily unlike the case of a fully solid-state system.

*5.2. Back-Reaction (2 H₂+O₂ → 2 H₂O)*

In 1985, Sato and co-workers noted that rate of the overall water splitting reaction over $Rh/TiO_2$ $Pt/TiO_2$ and $Pd/TiO_2$ catalysts declined with the accumulation of the products. They attributed this decline to the thermal back-reaction assisted by loaded metals with a reaction rate proportional to hydrogen pressure. The first-order rate constant of the back-reaction was found to be the lowest for catalyst with Rh loading [22]. Earlier, others using an $NiO/SrTiO_3$ photocatalyst found that the rate of reaction was approximately proportional to water-vapor pressure up to 10 Torr indicating a negligible thermal back-reaction [35]. These are probably a few of the reasons behind the relatively high activity of Rh and NiO as the reduction co-catalyst during numerous water splitting reaction

studies [59,61,62]. However, during hydrogen generation using sacrificial agents, Pt, Pd and Au are the preferred choice mainly due to their electron trapping and hydrogen atom recombination ability. Their activity towards back-reaction is ignored due to the absence of oxygen formation under such conditions [9,29,31,33,83,84,123]. In 2000, Anpo and co-workers investigated the back-oxidation reaction on $Pt/TiO_2$ systems under dark conditions and observed an increased reaction rate with increasing Pt loading [133]. There are few other accounts of back-reactions in the literature [76,134,135].

Later on, it was shown that the $Cr_2O_3$ coating of Rh metal to make a core-shell structure increases the rate of reaction on an Rh/GaN:ZnO photocatalyst (see Figure 5 for details) [79]. The core-shell structure is achieved by drying Cr and Rh metals deposited on the photocatalyst at 343K in air. The metal deposition is achieved in two steps by photodepositing Rh and Cr metal one after the other from their aqueous ionic solutions. In the second step, Cr prefers to sit on the Rh sites due to higher electronic density. The same effect was also noted for $Cr_2O_3$-coated Pt/GaN:ZnO [136]. A hydrated $Cr_2O_3$ nanolayer prevents back-reaction by selectively allowing protons to permeate for hydrogen evolution reaction [137], while hindering oxygen permeation to inhibit its reduction on metal sites [136]. Coatings of the whole photocatalyst instead of the cocatalyst by oxyhydroxides of Ti, Nb, and Ta have also been reported to suppress back-oxidation reactions [37,54,63,138]. Co-catalyst coverage with a suitable ionic or molecular adsorbate is another strategy to curb recombination reaction. For example, the presence of $I^-$ or $F^-$ anion in the reaction medium in the case of Pt-loaded $TiO_2$ photocatalysts results in lowering the rate of reverse reaction [76,139].

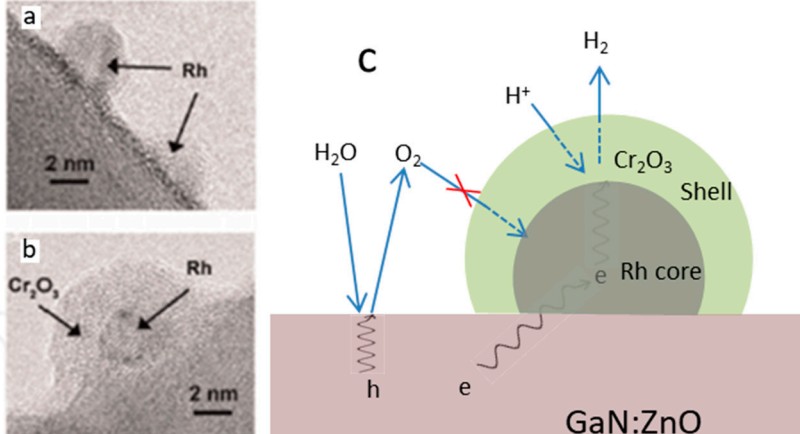

**Figure 5.** TEM images of GaN:ZnO after the photodeposition of (**a**) Rh and (**b**) $Rh/Cr_2O_3$ core-shell structure obtained by the photodeposition of Cr on Rh/GaN:ZnO followed by heat treatment at 343 K in air. Adapted with permission from reference [79]. Copyright © (2007), American Chemical Society. (**c**) Schematic representation of the role of the core-shell structure in the water splitting reaction indicating the inhibition of oxygen permeation to the Rh core and at the same time allowing $H^+$ to diffuse through the $Cr_2O_3$ shell, thus preventing back reaction. Reproduced with permission from reference [138]. Copyright © (2015), American Chemical Society.

In addition, the chemisorption of CO on Rh, Pt and Pd cocatalysts has been found to inhibit recombination reaction while maintaining the rate of hydrogen evolution. However, a CO oxidation side reaction also takes place at the same time [135]. The spatial separation of oxidation and reduction sites may also decrease the extent of crossover to a certain degree. For example, the photocatalytic activity of $NiO/NaTaO_3$ was remarkably increased by La doping partly due to the creation of a characteristic step structure at the crystal surface. The preferential NiO or Pt loading at the step edges effectively separated reduction from oxidation sites, where oxidation occurred inside the grooves [67,68].

### 5.3. Mass Transfer Limitations

Most photocatalytic overall water splitting reactions have been performed at the liquid-solid interface in the slurry phase, while a few have been carried at the gas-solid interface [16,18,79,140,141]. A mass transfer limitation especially in the slurry systems is the most overlooked yet important aspect in the photocatalytic water splitting research that can considerably affect the efficiency of the overall process. In such a system, the diffusion of gases from the catalyst surface to the gas phase is largely dependent on the reactor type, reaction media and stirring rate. The photocatalytic experimental parameters are not standardized in the literature unlike in the case of solar cells, therefore, it is difficult to compare the photocatalytic activity, apart from mass transfer limitations among different reported studies [142]. The photocatalytic activity of different systems may be compared using apparent quantum yield; however, it is often measured at different incident radiation intensities and wavelength ranges for similar catalytic materials that complicate the comparison [65,66].

For slurry phase reactions, the presence of various interfaces, as shown in Figure 6, may impose a non-negligible mass transfer limitation effect on the overall observed rates. Further, mass transfer limitations can affect gas generation rates by promoting back-reaction in two ways. (1) The rates of products transfer from the catalyst surface are slower than the reduction/oxidation reaction rates. (2) The apparent hydrogen and oxygen evolution rates at the liquid-gas interface are limited by the mass transfer rates. An enhancement in the hydrogen generation rate with an increase in stirring rate-indicating improvement in mass transfer at the gas–liquid interface has been demonstrated on $Pt/TiO_2$ photocatalyst [143]. An increase in the hydrogen production rate of ca. 1.4 times with an increase in the stirring rate from 350 rpm to 900 rpm in a water-ethanol mixture was noted [143]. Furthermore, at a fixed catalyst concentration a decrease in the rate per gram of catalyst with an increase in liquid volume indicated significant mass transfer limitations [143]. In another photocatalytic hydrogen production study over $Pt/g$-$C_3N_4$ and $NaTaO_3$:La in a fluidized reactor a 3.7 times enhancement in the activity at a speed from 0 rpm to 1035 rpm was observed [144].

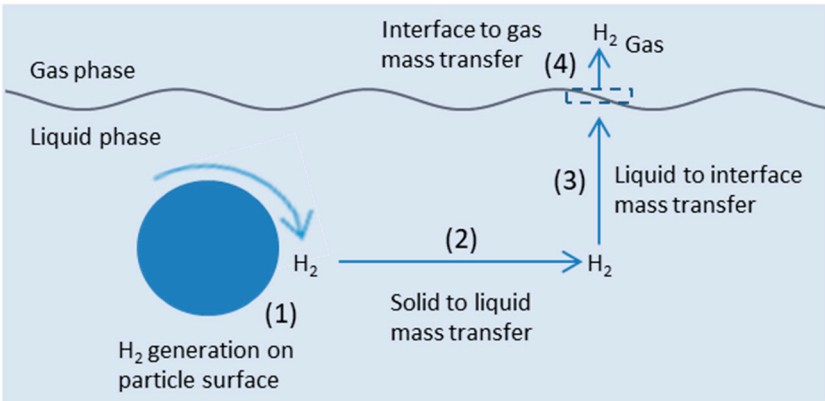

**Figure 6.** Different hydrogen mass transfer steps from its generation at the catalyst surface to the gas phase: (1) hydrogen production at the catalyst particle surface followed by its (2) transfer from the catalyst surface to the liquid phase, (3) diffusion to the liquid-gas interface and (4) evolution to the gas phase at the liquid-gas interface. Reproduced with permission from reference [145]. Copyright © (2018), Elsevier.

In another study, a mathematical model was developed to fit experimentally measured photocatalyst performances of $Pt/TiO_2$ (Degussa P25) in a fluidized bed reactor. This allowed some parameters such as the back-reaction rate constant, the mass transfer coefficient and photon extinction coefficient to be estimated, and the maximum achievable hydrogen production rate to be predicted. It was found that the hydrogen production rate and the apparent quantum yield of the system were directly controlled by the rate of mass

transfer in the reactor. A 350% increase in the hydrogen evolution rate was achieved when the rate of mass transfer was made significantly large with respect to the back-reaction [145]. In the case of immobilized catalyst, there are two types of mass transfer limitations; external mass transfer and internal mass transfer. While the former could be reduced to a negligible value by increasing the flow rate (Reynolds number) of the solution, the latter is an intrinsic property of the catalyst film, which is difficult to alter [146]. The Reynolds number is a ratio of inertial to that of viscous forces within a fluid. At its low values, flow tends to be laminar (dominated by viscous forces), while at high values the flow tends to be turbulent (dominated by inertial forces). From the above discussion, it appears that further work on the mass transfer dependence of hydrogen generation rates in different water splitting reaction experiments is needed. These studies may help to enhance the rates and lead to a better fundamental understanding of the photocatalytic reaction steps.

## 6. More Recent Overall Water Splitting Systems

Recently, a few more advanced (by design) overall water splitting-systems have emerged. A brief account of some of them is given next. One example is a sheet system based on a solid-state electron mediator in which a mixture of reduction and oxidation catalyst particles is embedded in a thin conductive layer such that a sufficient contact area for efficient charge transfer is achieved [16–19]. Typical example includes Ru-loaded $SrTiO_3$:La, Rh as a reduction catalyst and $BiVO_4$:Mo as an oxidation catalyst put on either an Au or carbon conductive layer (see Figure 7). In the case of carbon, this assembly exhibited an STH of 1.2%, which is to be compared to that of 0.2% for Au [16,17]. Apparently, a six times higher efficiency in the former case was attributed to the presence of the $Cr_2O_3$ layer and lower activity of the carbon conductive layer for back-reaction as compared to the Au layer. This technique does not necessarily require an electrolyte and the photocatalysts can be easily replaced when required. Both these characteristics associated to this unique system are advantageous to the design of practical solar water splitting units.

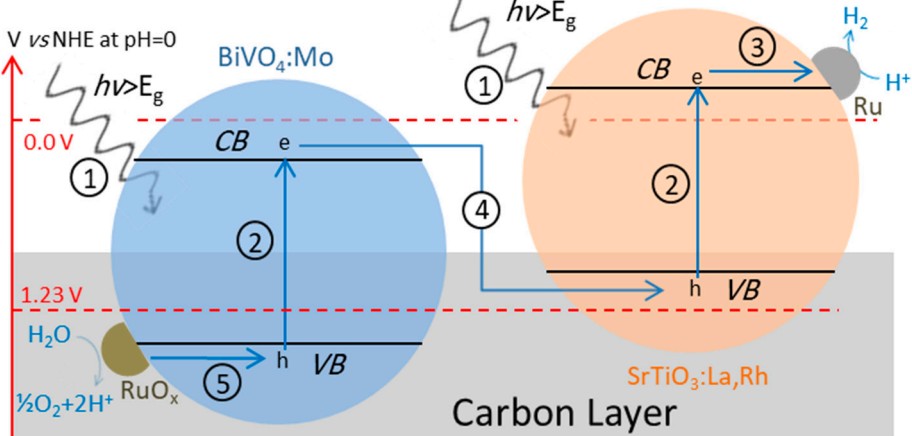

**Figure 7.** Z-scheme using a carbon sheet as an electron transfer medium: (1) light absorption, (2) electron excitation from the VB to the CB, (3) electron(s) transfer to Ru metal cocatalyst for $H^+$ reduction to produce $H_2$, (4) electron(s) transfer from the CB of an oxidation catalyst to the VB of reduction catalyst via a conductor carbon layer (electron-hole recombination) and (5) hole transfer to $RuO_x$ cocatalyst to catalyze water oxidation Reproduced with permission from reference [16]. Copyright © (2007), American Chemical Society.

One of the drawbacks of using these conductive layers is their deposition using a costly sputtering process, which is not promising for large-scale fabrication. Alternatively, these layers can be made from other less expensive conductive materials such as reduced graphene oxide [18]. The photocatalysts particles can be deposited on conductive layer electrically or using a printing technique with an ink containing catalyst powders [141]. However, such systems have shown far lower performance than the former cases, and hence

further studies and improvements are required. The working principle of the photocatalyst sheet is similar to that for a p–n tandem cell in which a $SrTiO_3$:Rh, La photocathode and a $BiVO_4$:Mo photoanode are connected in a parallel configuration. The performance of such sheets is found to be superior to tandem photoelectrochemical water splitting primarily due to the close proximity of reduction and oxidation reaction sites, which lowered the solution resistance [147]. Another direction to study overall water splitting uses metals such as Au and Ag because of their surface plasmon resonance (SPR) properties. For example, one characteristics of plasmonic enhanced photocatalysts is that the absorption of sunlight can be expanded to almost all the solar spectral range depending on the type of metal used and its particle size distribution [148,149]. Moreover, these metals also play cocatalyst function at the same time [29,30,140,150,151]. It was noted that the hydrogen production rate observed under UV light exposure increases with a simultaneous exposure to visible light in the presence of Au and Pd supported on $TiO_2$ and organic sacrificial agents [30,150] however no such effect was seen in the case of Ag when supported on $TiO_2$ [29].

Another important work in this direction was reported by Moskovit and co-workers who demonstrated an Au nanorods-based autonomous overall water splitting device in which all charge carriers originated from the generation of hot electrons resulting from the excitation of surface plasmons [152]. $TiO_2$ was deposited on one end of each nanorod to act as an electron filter. Pt loaded on an electron filter layer and Co deposited directly on the bare nanorod were used as reduction and oxidation catalysts, respectively (see reference 156 for further details). They reported approximately 1.14 mmol $h^{-1}$ $g^{-1}$ per gram of (active photocatalytic device weight; Au, Pt, $TiO_2$, Co) hydrogen production rate under one sun illumination with long-term operational stability. The biggest drawback of making use of plasmonic response is the capex cost of the plant at the commercial scale as this response is only shown by expensive noble metals. Another work has demonstrated overall water splitting by decorating CdS nanorods with a Pt reduction and a molecular oxidation catalyst (a ruthenium organometallic complex (Figure 2), with dithiocarbamate groups for chemical ligation). The process relies on the nanorod morphology of CdS to spatially separate the reduction and oxidation sites [153]. However, given the photo-corrosion of CdS even in the presence of sacrificial agents, and the fact that the rate of oxygen production went to zero after about 100 min indicate that the absence of a catalytic reaction and therefore the absence of water splitting. Recently, Takata and co-workers demonstrated a system comprised of $Rh/Cr_2O_3$ as hydrogen and CoOOH as an oxygen evolution cocatalyst both loaded on Al doped $SrTiO_3$ photocatalyst [15,154]. The catalyst exhibited an external quantum efficiency reaching 96% and an internal quantum efficiency reaching unity in a 350–360 nm irradiation wavelength range. The activity of the photocatalyst was thought to mainly originate from two factors: the selective photo-deposition of Rh and Co on different crystal facets of the semiconductor particles resulting in an anisotropic charge transport, and the fact that (or as a consequence of) molecular hydrogen and oxygen productions were spatially separated avoiding the back-reaction [155].

## 7. Reactor Design and Cost of Hydrogen

Over the years, the primary motive behind working on colloidal particle photocatalytic systems was the conceptual simplicity and their possible low cost. Despite many decades of technical research, there is no unbiased analysis available to determine whether research dollars spent on such systems will lead to commercially viable solutions. The first detailed technoeconomic analysis was published in 2009, which was a study conducted by the United States Department of Energy (DOE) on hydrogen generation based on different conceptual systems [11]. This was later summarized by others in a different study [13]. The two types of colloidal systems (Figure 8) for water splitting considered in the study were:

Type-1: A single electrolyte filled reactor bed containing a colloidal suspension, which produced a mixture of hydrogen and oxygen product gases.

Type-2: Dual electrolyte-filled reactor beds containing colloidal suspensions, with one bed carrying water oxidation and the other bed carrying water reduction and including a mechanism for circulating the ions between the beds.

The two systems utilize aqueous reactor beds containing colloidal suspensions of photocatalytic nanoparticles to generate hydrogen and oxygen, a gas processing system that compresses and purifies the output gas stream, and ancillary equipment. The systems were sized for an average daily production of 1000 kg hydrogen, operating at a STH of ~5% and 10%, respectively, and they had compression units to get hydrogen at 305 psi [13]. The base case results of this study placed the levelized cost of hydrogen at 1.60 $/kg and 3.2 $/kg, for type 1 and 2 systems, respectively (Figure 9) [13]. The capital costs were the major contributor to the calculated cost of hydrogen where gas processing dominated for the type 1 system and the reactor cost dominated for the type 2 system. These numbers suggested that commercial-scale photocatalytic water splitting could be cost-competitive with fossil-based fuels and spurred research in this direction. However, over the last decade many researchers across the world have realized that achieving the assumptions made in this study is not possible at present nor in the foreseen future. After 50 years of research, the best particle systems have reached ca. 1% STH with limited stability (results still needing confirmation by others) [141]. Thus, achieving 10% and 5% STH with stability for more than 5 years is far away. In addition, the cost and challenges along with safety concerns associated with separating $H_2/O_2$, which are produced together in type 1 reactor, were hugely underestimated. The technoeconomic analysis assumes pressure swing adsorption (PSA) for $H_2/O_2$ separation. While PSA plants can be used to recover and purify hydrogen from hydrogen-rich streams such as synthesis gases resulting from steam reforming, or to separate oxygen and nitrogen from air, the separation of $H_2/O_2$ is still not possible using them.

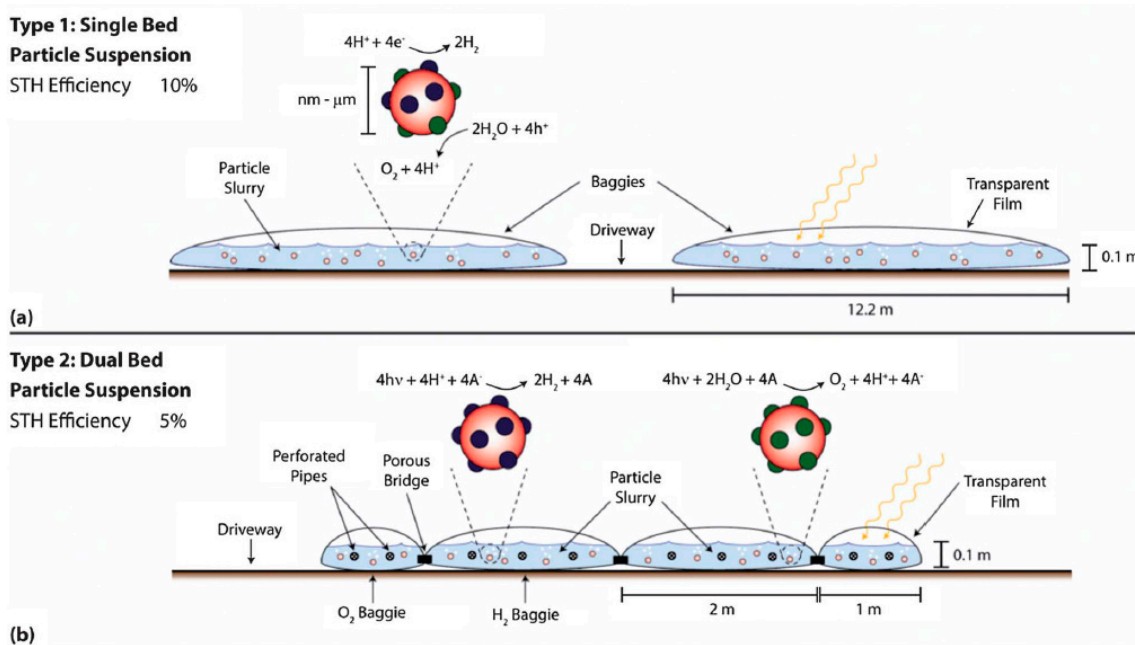

**Figure 8.** Design of hydrogen production reactors based on colloidal catalyst suspensions. (**a**) Type 1 reactor cross-section showing the particle slurry contained within baggies separated by an access driveway. A photocatalyst suspension capable of overall water splitting is required. (**b**) Type 2 reactor cross-section showing the particle slurries contained within baggie assemblies consisting of an alternating arrangement of a full size and half-size baggie each for hydrogen and oxygen evolution. This design requires the use of a redox mediator ($A/A^-$) and porous bridges to transport it from one compartment to the other. Perforated pipes running the length of the baggies facilitate mixing of the redox mediator. Adapted with permission from references [11,13]. Copyright © (2008), Royal Society of Chemistry.

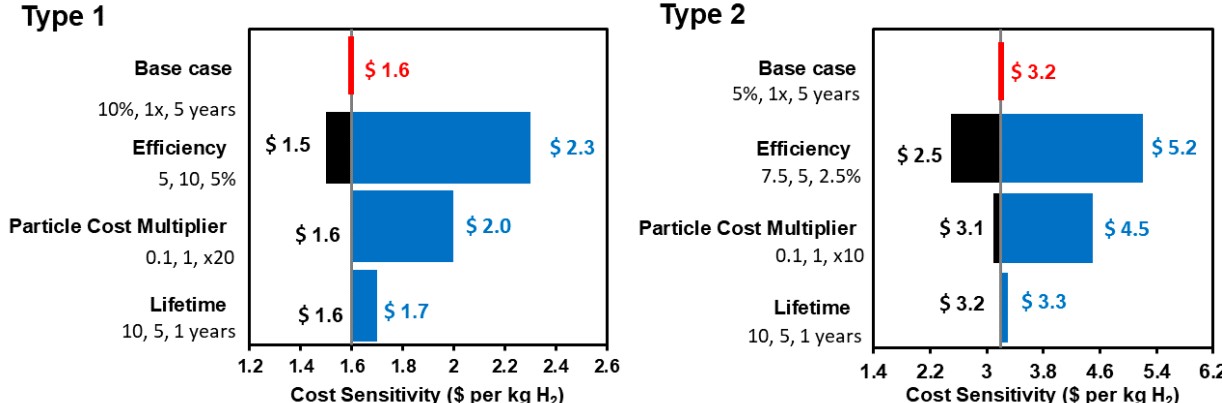

**Figure 9.** Effect of efficiency, particle or panel cost, and component lifetime on the cost of hydrogen from the two reactors. Each calculation represents the variation of a single parameter as indicated on the left axis from the base case to a higher and lower value. Reproduced with permission from reference [13]. Copyright © (2008), Royal Society of Chemistry.

To overcome this, others suggested using $N_2$ and $CO_2$ as diluents to eliminate the explosion hazard but this comes with an increase in cost [156,157]. Using $N_2$ as a diluent, a membrane cascade design was proposed to recover the hydrogen outside the flammability range. Two different tertiary ($H_2$, $O_2$ and $N_2$) mixtures with 4 and 6 mole % hydrogen were investigated to recover hydrogen using a commercial polyimide membrane. The results showed that 92% recovery and 92.5% purity was possible for a 4 mole % mixture at a specific recovery cost of 8.20 $/kg. On the other hand, for 6 mole % mixture, 90% recovery and 95% purity were achievable with a specific recovery cost of 6.15 $/kg [156]. The reduction in recovery cost is the result of the lower feed flow rates in the latter case with higher hydrogen concentration. These two factors play a major role in reducing the required membrane area and the compression power. The utilization of $CO_2$ as a flammability suppressant was found to be more effective than $N_2$. A higher purity product and lower specific cost with an overall slight decrease in recovery rate can be achieved with the $CO_2$ diluent system. The best economic results for this process were obtained at an 85% recovery rate with 99% product purity at a cost of 6.40 $/kg [157].

Another design based on a fixed panel array, which generates molecular hydrogen, and oxygen separately has been discussed in the 2009 DOE report and later by Shaner and coworkers in 2016. See Figure 10 [11,12]. This design is primarily based on using photovoltaic cells but in principal could also be used for colloidal photocatalyst materials. The photocatalysts can be coated on conducting electrodes, which are connected electrically, and $H_2/O_2$ can be produced separately. While this design addresses the challenges associated with separation, it adds more area, cost and complexity to the reactor system. At the same time, decoupling of the absorber and catalyst areas presents additional design challenges and costs. Given the large scale of all the reactors, there may be a significant voltage loss due to the voltage drop across the solution. The calculated levelized cost of hydrogen for such systems was reported in the range of 10–12 $/kg. Capital costs dominate the price per kg of hydrogen, with more than 80% of the cost originating from the materials, construction, and installation of the photoactive cells. Given the rigid encapsulation framework, as opposed to the flexible baggies of the Type 1 and 2 reactors, compression cost for this type of reactor is also high since the auxiliary units must be sized for the peak hourly output and not the average output over the day. The cost of hydrogen produced from such a system could be lowered with the use of photocatalysts powder materials coated on conducting electrodes instead of PV.

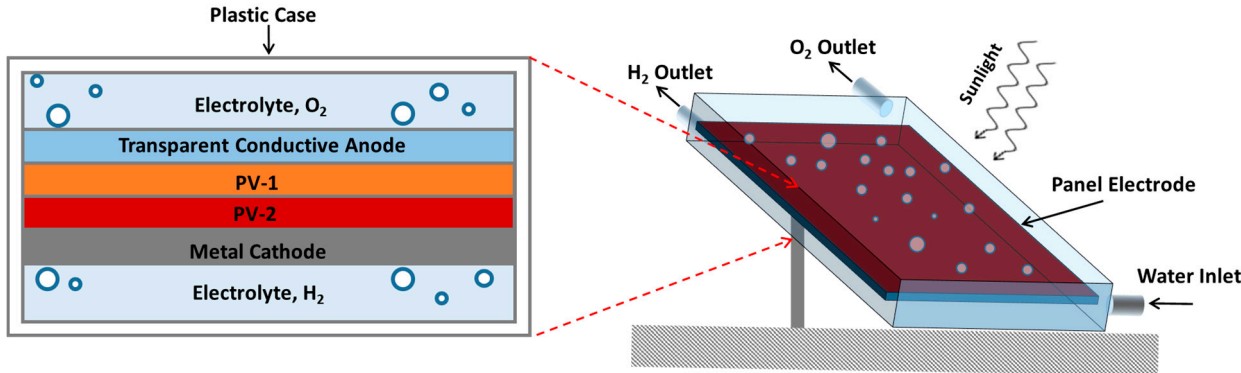

**Figure 10.** Fixed panel array for spatially separated hydrogen and oxygen production. The panel is comprised of two photoactive layers sandwiched between a transparent conductive anode and a metal cathode. While the former catalyzes an oxygen evolution reaction and allows photons to pass through, the latter is responsible for a hydrogen evolution reaction. The whole panel is encased in a transparent plastic layer and is tilted in a way to see maximum light from the sun. Reproduced with permission from reference [11,12]. Copyright © (2008), Royal Society of Chemistry.

## 8. Conclusions and Future Perspective

The main underlying principles related to the overall water splitting using particulate photocatalysts are summarized. Various types of semiconductors including oxides, nitrides and their derivative oxynitrides have been developed over the past three decades. Some visible light responsive (up to around 500 nm; $E_g$ = 2.5 eV) photocatalysts have been demonstrated [16,158]. In order to become competitive with conventional industrial processes such as steam methane reforming an STH around 10% is required as discussed under the reactor design and cost of the hydrogen section [13,14]. To meet this goal a photocatalyst capable of splitting water through one-step excitation must have absorption edge between 350 and 1000 nm. The shorter wavelength bound is due to the fraction of total solar incident light flux that is needed to be converted and the longer wavelength bound is due to the minimum $\Delta G$ (1.23 V $\approx$ 1000 nm) requirement to split water. Although one would intuitively prefer to have a photocatalyst with band gap as close as possible to the longer wavelength bound, energy losses during the reduction and oxidation reactions are unavoidable; these can be up to 0.5 V.

While most of the oxide catalysts absorb UV light, a range of (oxy)nitride and oxysulfide semiconductors have an absorption edge close to 600 nm indicating their potential as candidates to achieve the target efficiency [37,125,159]. Thermal and photostability are chronic issues of all nitride-based catalysts that need to be solved. One way to achieve this was through the use of cocatalysts, with some modest progress [37,159–161]. Enhanced light harvesting is also found for some plasmonic photocatalysts [152,162]. This concept may become useful for water splitting once with better fundamental knowledge is assimilated within the catalysis community. In the case of the Z-scheme systems, all three cocatalyst-semiconductor, cocatalyst-solution and semiconductor-solution interfaces must be rationally designed after considering their, optical, electronic and catalytic properties to maximize forward electron transfer rates. Furthermore, designing new integrated systems based on rational interface-engineering strategies is another approach that can make use of the inherent properties of semiconductor materials. An example of this in photocatalyst sheet systems is given [17].

More efforts directed towards understanding the fundamental photocatalytic processes, in particular charge carrier dynamics, using advanced characterization techniques are poised to open new opportunities to enhance performance [122,163]; although for this to be useful materials need to be characterized at the atomic scale and reproducibly made by many other groups. Another important challenge for industrial-scale application of such systems is to eliminate, or at least considerably decrease the use of rare, expensive or polluting elements. The separation of explosive $H_2$–$O_2$ mixtures and related safety issues is another important hurdle that need to be crossed [156]. Given our current understanding

of the overall system, the development of a visible light responsive stable photocatalyst dispersed in water in the form of particles that also meets a 10% STH requirement is still very challenging at both the scientific and engineering levels.

**Author Contributions:** M.A.N., M.A.K. and A.A.Z.: preparation of first draft; M.A.N. and H.I.: revisions and discussions on subsequent drafts. All authors have read and agreed to the published version of the manuscript.

**Funding:** The manuscript was written using resources available at [1]SABIC Technology Center, King Abdullah University of Science and Technology, Thuwal 23955, Saudi Arabia.

**Conflicts of Interest:** The authors declare no conflict of interest.

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
