# Peer review of "An Overview of the Photocatalytic Water Splitting over Suspended Particles"

_catalysts, doi:10.3390/catal11010060_

Round 1

Reviewer 1 Report

In this manuscript, the authors gave an overview of the photocatalytic water splitting over suspended particles. This manuscript provides essential information for an overview-type review article, but still, the authors need to address some of the major issues present in the manuscript. There are a lot of formatting and grammatical errors can be found in the manuscript, and the English writing and figure quality should be polished more.

Comments

  1. Abstract of the manuscript is written irregularly, the authors should rewrite the abstract with the proper grammatical format. Providing a readable abstract will be helpful for the readers and improve the impression of the manuscript.
  2. There are some formatting errors in the manuscript, in page 3 the references are given in the subscript “[17,40-48]”, it should be given in an appropriate format. Like this, there are a lot of reference formatting errors in the manuscript.
  3. On page 4, the explanation for co-catalyst is not mentioned, a brief introduction about co-catalyst should be given. The following sentence should be given meaningfully. “This is commonly called “co-catalyst” to help improve the kinetics of the reaction.”
  4. The image is given in Figure 2 is unclear and incomprehensible, the quality of the figures should significantly be improved. The caption of figure 2 is not well presented, it can be given as “Schematic representation of major synthesis methods of photocatalyst for overall water splitting applications”.
  5. On page 5, the sentence “As for any heterogeneous catalysis material the design of these multicomponent catalysts needs attention in particular in terms of their energy levels and electronic structures” should be corrected to give appropriate meaning.
  6. On page 5, authors stated that “Transition metals (in particular noble metals) such as Pt [24,25], Rh [22], Ru [82], Au [83,84], Cu [85] and Ni [23]”, it is better to mention precious and non-precious transition metals separately (Cu and Ni).
  7. On page 6, the authors mentioned that “UV portion of the solar spectrum is up to 4% or so only” since the authors addressed this as the limiting factor for oxide catalyst, proper references for this data should be provided.
  8. The authors failed to give a chronological order for the subheadings in this manuscript. On page 7, “3. Z-scheme: A two-step approach” and “4.3.1. Z-scheme with aqueous redox mediator” is not appropriate. All the subheadings should be checked once again and should be given in proper order.
  9. Quality of figure 3 – Figure 11 should be significantly improved, the reaction mechanism of H2 and O2 conversion given in figure 3 is not properly visible. In the caption of Figure 3, what does the author convey through the text “[more details]”? 
  10. It is obvious on page 8 that there was a mismatch in letters font size in the texts “In general a pH adjustment is required to stabilize the redox couple. For example, I-/I3- is stable
    in acidic where IO3-/I- is stable in basic aqueous media”. Also, there are lots of font errors can be found, the authors should check carefully and correct the concerns in text font size.
  11. The caption for Figure 5 is too long, some of the explanations given in the captions can be transferred to the main text.

Author Response

Please see the response to the referee v1

Reviewer 2 Report

1) Figure 1 has a black half circle near the hydrogen reduction equation. Please remove this.

2) Figure 2 is too wordy. Can it be changed to a table instead? It looks like one.

3) Line 191. Since as a Ni. What does this mean? Change the wording.

4) Line 234. "Yet, again it is unclear if the reaction is indeed catalytic for these type of catalysts" Change the wording.

5) Line 365. Nature.

6)  Line 406. Back reaction or Reverse Reaction?. Reverse reaction sounds better in terms of chemical equilibrium.

7) Lines 677-679 are incomplete.

Author Response

Please see response to version v1.

Round 2

Reviewer 1 Report

The authors have addressed most of the comments raised. The quality of images is much improved and the whole manuscript has been improved as well. Thus, I recommend the manuscript can be accepted for publication in this current form in Catalysts.